# Self-Powered Sb_2_Te_3_/MoS_2_ Heterojunction Broadband Photodetector on Flexible Substrate from Visible to Near Infrared

**DOI:** 10.3390/nano13131973

**Published:** 2023-06-29

**Authors:** Hao Wang, Chaobo Dong, Yaliang Gui, Jiachi Ye, Salem Altaleb, Martin Thomaschewski, Behrouz Movahhed Nouri, Chandraman Patil, Hamed Dalir, Volker J. Sorger

**Affiliations:** 1Optelligence LLC, 10703 Marlboro Pike, Upper Marlboro, MD 20772, USA; hwang40@gwu.edu (H.W.);; 2Department of Electrical & Computer Engineering, University of Florida, 968 Center Drive 216 Larsen Hall, Gainesville, FL 32611, USA; alcatraz@gwu.edu (J.Y.); hamed.dalir@ufl.edu (H.D.); 3Department of Electrical and Computer Engineering, The George Washington University, 800 22nd Street, Washington, DC 20052, USA; chaobo17@email.gwu.edu (C.D.); ygui82@gwu.edu (Y.G.); thomaschewski@gwu.edu (M.T.);

**Keywords:** 2D materials, photodetector, broadband, self-powered, flexible substrate, photonic integrated circuits

## Abstract

Van der Waals (vdWs) heterostructures, assembled by stacking of two-dimensional (2D) crystal layers, have emerged as a promising new material system for high-performance optoelectronic applications, such as thin film transistors, photodetectors, and light-emitters. In this study, we showcase an innovative device that leverages strain-tuning capabilities, utilizing a MoS_2_/Sb_2_Te_3_ vdWs p-n heterojunction architecture designed explicitly for photodetection across the visible to near-infrared spectrum. These heterojunction devices provide ultra-low dark currents as small as 4.3 pA, a robust photoresponsivity of 0.12 A W^−1^, and reasonable response times characterized by rising and falling durations of 0.197 s and 0.138 s, respectively. These novel devices exhibit remarkable tunability under the application of compressive strain up to 0.3%. The introduction of strain at the heterojunction interface influences the bandgap of the materials, resulting in a significant alteration of the heterojunction’s band structure. This subsequently shifts the detector’s optical absorption properties. The proposed strategy of strain-induced engineering of the stacked 2D crystal materials allows the tuning of the electronic and optical properties of the device. Such a technique enables fine-tuning of the optoelectronic performance of vdWs devices, paving the way for tunable high-performance, low-power consumption applications. This development also holds significant potential for applications in wearable sensor technology and flexible electro-optic circuits.

## 1. Introduction

Photodetectors are a crucial component in a wide range of applications, such as optical communication, sensing, imaging, and energy harvesting [1,2,3]. In recent years, there has been a growing interest in developing high-performance, flexible, and self-powered photodetectors for use in wearable intelligent sensors. Flexible photodetectors are designed to be bendable and conform to various shapes and curvatures, making them suitable for use in unconventional form factors, such as smart textiles, medical devices, and electronic skin [4,5]. These devices have a wide range of potential applications, including health monitoring, military surveillance, and flexible displays [6,7,8,9]. The development of flexible photodetectors requires the use of novel materials and device architectures that can withstand bending and stretching without compromising their electrical properties. Polymer films such as polyimide (PI), polyethylene naphthalate (PEN), or polyethylene terephthalate (PET) are common substrates used for flexible optoelectronics. Their mechanical properties, such as Young’s modulus and elastic strain, are in the range of 2–7 GPa and 1–3.5%, respectively [10,11,12]. Self-powered photodetectors, also known as photovoltaic photodetectors, can operate without external power sources. These devices work by converting light energy into electrical energy through the photovoltaic effect. Recent studies have shown that self-powered photodetectors based on 2D materials have several advantages over conventional device configurations, including higher sensitivity, lower power consumption, and smaller size, making them an attractive device platform for future optoelectronic technology [13,14]. 

Transition metal dichalcogenides (TMDs) have been widely explored as active materials for high-performance optoelectronic devices due to their unique electronic and optical properties [15,16,17]. TMDs are layered materials consisting of two-dimensional (2D) layers of transition metal atoms (such as Mo, W, and Ti) sandwiched between chalcogen atoms (such as S, Se, and Te) in a hexagonal lattice structure [18]. The layered crystal structure results in unique electronic and optical properties that enable promising new applications. One of the critical properties of TMDs, such as a monolayer of MoTe_2_, WSe_2_, and WS_2_, that makes them attractive for optoelectronics is their direct bandgap in the visible to the near-infrared (NIR) range [19,20]. This direct bandgap allows TMDs to absorb light efficiently, making them suitable as active materials for photodetection. Another essential property of TMDs for optoelectronics is their high electron mobility, which enables fast charge transport and efficient device performance. The electron mobility in TMDs can range from a few hundred to several thousand cm^2^V^−1^s^−1^. The hole mobility in TMDs can range from a few tens to several hundred cm^2^V^−1^s^−1^ [20,21,22]. For example, MoS_2_ (molybdenum disulfide), which is one of the most studied TMDs, has been shown to have electron mobility values in the range of 10–10^3^ cm^2^V^−1^s^−1^ and hole mobility values in the range of 20–200 cm^2^V^−1^s^−1^ [21,23]. TMDs also exhibit significant nonlinear coefficients due to their 2D layered structure, which enhances their nonlinear light-matter interactions, including photoluminescence and Raman scattering [24,25,26]. In addition, the atomically thin nature of TMDs allows for the fabrication of flexible and transparent devices, making them suitable for wearable and flexible optoelectronics [10,27,28]. However, TMDs are frequently impeded by surface oxidation, high contact resistance, and low mobilities [29,30,31]. To overcome these shortcomings in TMD-based photodetectors, recent advancements have been made utilizing techniques such as plasmonics to enhance the light-matter interaction of TMDs and heterostructures to modify their intrinsic electrical and optical properties as the built-in electric field can efficiently separate the electron-hole pairs. These methods have demonstrated significant performance improvements, including improved sensitivity, broader frequency response, enhanced external quantum efficiencies (EQE), heightened detectivity (D*), and a more extensive spectral photoresponse [20,32,33,34]. The combination of transition metal dichalcogenides with other emerging materials, such as topological insulators (TIs), has not been widely explored yet. TIs possess several attractive optoelectronic properties, such as a unique energy band structure or topologically protected conductive edge or surface states, which offer higher mobility and a broader detection spectrum than graphene, which lacks a bandgap. The low thermal conductivity makes it beneficial to reduce heat dissipation and improve the efficiency of photovoltaic devices. In addition, TIs are topologically protected, which means that their surface states are robust against perturbations and immune to scattering by non-magnetic impurities or defects. [35,36,37,38,39] This makes TIs highly stable and durable, which is vital for the development of practical optoelectronics devices. 

Molybdenum disulfide (MoS_2_), as one of the most representative TMDs, has a direct bandgap of 1.8 eV, whereas, in bulk, it has an indirect bandgap of 1.3 eV. It can be used in various optoelectronic applications, such as photovoltaics and photodetection due to its unique optical and electronic properties, including a direct bandgap, high absorption coefficient, and absorption spectrum extending from visible to infrared [40,41,42,43,44]. Antimony telluride (Sb_2_Te_3_) is a topological insulator. Its bandgap ranges from approximately 0.2 to 0.3 eV, which places it in the category of a narrow-bandgap semiconductor. It exhibits a topologically protected surface state that can lead to high carrier mobility and low noise [36,38,45]. Heterojunctions are widely applied for self-powered photodetectors due to the built-in electric field that can efficiently separate the electron-hole pairs to enhance the photoresponsivity and speed under zero bias. By combining Sb_2_Te_3_ and MoS_2_ in a heterojunction structure, it is possible to achieve complementary optoelectronic properties and enhance the performance of the resulting photodetector. Previous studies have demonstrated the potential of Sb_2_Te_3_/MoS_2_ heterojunction-based photodetectors. For example, Wang et al. demonstrated a self-powered photodetector based on a Sb_2_Te_3_/MoS_2_ heterojunction that covered a wavelength range from visible to near-infrared (NIR). The device exhibits a low dark current of 2.4 pA at zero bias and high photoresponsivity of >150 mA W^−1^ at zero bias [41]. Liu et al. reported a photodetector based on a Sb_2_Te_3_/MoS_2_ heterojunction that exhibits a high photoresponsivity of 130 A W^−1^ and a detectivity of 2.2 × 10^12^ Jones [46]. However, these previous studies did not address the challenges of developing flexible and self-powered Sb_2_Te_3_/MoS_2_ heterojunction-based photodetectors, as the structural flexibility of the device is critical for applications, such as wearable electronics and conformal sensors. The exceptional performance of a photodetector is significantly underlined by its capacity to detect a broad spectrum of wavelengths. Recent research has demonstrated such capabilities in photodetectors through innovative designs [47]. For instance, in a study conducted by Yao et al., the researchers unveiled a photodetector leveraging the unique properties of a self-powered PbS/TiS_3_ heterojunction. This photodetector exhibited the capability to operate across an broad spectrum, ranging from the visible light to the near-infrared (NIR) range. Impressively, the device demonstrated a high responsivity of 0.36 A W^−1^, indicating a promising path for future research [48]. In a separate study, Talib et al. presented a highly sensitive photodetection solution using horizontally aligned titanium disulfide nanosheets (TNs). This inventive photodetector showcased the ability to detect light across an even broader spectrum, encompassing ultraviolet (UV), visible, and near-infrared wavelengths [49]. These developments are noteworthy as they show how advancements in heterostructures and nanomaterials can enhance the performance and functionality of photodetectors. This observation emphasizes the significance of designing photodetectors with broad detection capabilities. Here, we present the successful demonstration of a self-powered broadband photodetector incorporating a Sb_2_Te_3_/MoS_2_ heterojunction on a flexible PI substrate. We provide a thorough, extensive analysis of the impact of compressive strain on the device’s performance. This photodetector showcases remarkable performance characteristics, encompassing a wide detection wavelength range spanning from visible to NIR. Notably, it exhibits an exceptionally low dark current of only 4.3 pA, highlighting its excellent signal-to-noise ratio. Moreover, the device achieves a high photoresponsivity of 120 mA W^−1^, emphasizing its exceptional sensitivity to incoming light. Furthermore, when subjected to a compressive strain up to 0.3%, the photodetector exhibits remarkable stability, ensuring reliable performance even under challenging conditions.

## 2. Materials and Methods

The experimental procedures for testing the Sb_2_Te_3_/MoS_2_ PN junction heterostructure devices incorporated the utilization of key equipment for achieving precision and reliability in electrical response measurements. Firstly, we deployed an advanced tunable source, the NKT SUPERCONTINUUM Compact, for device experimentation. This tool served a critical role in our experimental setup, providing us with a robust and precise light source. Next, for the purpose of accurately gauging the electrical responses, we utilized a sophisticated source meter, the Keithly 2600B (Keithley Instruments, Cleveland, OH, USA). This instrument granted us the ability to measure the electrical properties of the tested devices with exceptional precision and reliability. The experimental process also incorporated a highly focused laser beam. To ensure an optimal focus on the devices under testing, we employed an objective lens. This precision tool allowed us to concentrate the laser beam precisely on the device, facilitating the optimal interaction necessary for the reliable measurement of electrical responses. Through this meticulously designed experimental setup, we were able to conduct a comprehensive analysis of the electrical characteristics of the Sb_2_Te_3_/MoS_2_ PN junction heterostructure devices.

### Fabrication Process

The construction of the Sb_2_Te_3_/MoS_2_ PN junction heterostructure involves a sophisticated and meticulously designed sequence of steps to ensure accurate fabrication and optimal device performance. The process commences with the derivation of thin flakes from bulk crystals of Sb_2_Te_3_ and MoS_2_. This crucial step employs a pick-and-drop transfer system, providing careful handling and precise placement of the thin flakes onto a Polyimide (PI) substrate. Following the substrate preparation, we proceed to establish the electrical contacts. This is performed using state-of-the-art electron beam lithography technology, specifically employing the Raith Pioneer EBL system. This high-resolution process allows us to achieve accurate and fine patterns necessary for the formation of electrical contacts. Subsequently, we form the metallic Ti/Au (5 nm/45 nm) electrodes. This stage incorporates the electron beam evaporation method, a proven technique for depositing thin film materials with high precision and uniformity. Finally, the process concludes with a lift-off procedure. Here, the excess metal from the deposition process is removed using acetone at room temperature. This is then followed by a rinsing process using isopropyl alcohol to ensure a clean and residue-free structure. The final step of the process involves drying the structure with nitrogen, ensuring a moisture-free and clean device that is ready for testing or integration into a larger system.

## 3. Results

The heterojunction of MoS_2_ and Sb_2_Te_3_ flakes is fabricated by mechanically exfoliating the flakes from their respective bulk crystals. The flakes are carefully transferred onto a clean, flexible polyimide thin film in a specific sequence using a 2D transfer system that minimizes contamination of material surfaces and avoids potential structural damage to the flakes. The device schematic is illustrated, providing a visual representation of the different components of the heterojunction and their arrangement (Figure 1a). The false colored scanning electron microscope image of the fabricated Sb_2_Te_3_/MoS_2_ is depicted in Figure 1b, where the Sb_2_Te_3_ flake is transferred atop of the MoS_2_ flake. The assembly of the Sb_2_Te_3_/MoS_2_ heterostructure involves a detailed fabrication process comprising several methodical steps. The process commences with the exfoliation of Sb_2_Te_3_ flakes from a bulk crystal using blue tape. These flakes are then meticulously relocated to the chosen substrate. In a parallel action, MoS_2_ flakes are detached from another bulk crystal using an identical exfoliation technique. These flakes are then carefully laid onto a polydimethylsiloxane (PDMS) film. In the concluding stage, the PDMS film bearing the MoS_2_ flakes is carefully aligned over the Sb_2_Te_3_ flakes. As a glass slide is slowly retracted from the surface, the van der Waals force that develops between the MoS_2_ and Sb_2_Te_3_ flakes surpasses the bonding force between the PDMS and MoS_2_. This shift in forces allows the Sb_2_Te_3_ flake to bond with the MoS_2_ flake, culminating in the formation of a heterostructure junction. This meticulous process is instrumental in creating the desired properties for optoelectronic applications. The thickness of MoS_2_ and Sb_2_Te_3_ are estimated to be around 25 nm and 65 nm, respectively, which are measured by Raman Spectra [41]. The electrodes, composed of Ti/Au (5/45 nm), were formed to establish electrical contacts with both the MoS_2_ and Sb_2_Te_3_ flakes. The deposition of the gold electrodes ensures a good electrical connection and minimizes parasitic effects that could affect the performance of the heterojunction for the chosen materials here. To study the impact of strain on the p-n heterojunction, the flexible device was bent upwards to apply tensile strains. The applied strain was calculated based on the introduced bending angles, which were carefully measured using a customized tensile stress setup. The application of tensile strains allows for the investigation of the impact of mechanical deformation on the electronic and optical properties of the heterojunction. Finally, the flexible heterojunction was characterized under different wavelengths and intensities of illumination using a supercontinuum laser source. The detailed characterization process includes measuring the electrical and optical properties of the heterojunction as a function of the applied strain. The results of this study could have significant implications for the development of flexible and high-performance electronic devices. 

Electron-hole pairs are generated at the junction upon incident optical illumination on the material, which can be measured as an electrical current under electrical bias. Nonetheless, a p-n heterojunction can even detect light in the absence of external bias voltage due to the presence of built-in junction potential. This phenomenon results from the alignment of energy levels in both materials (Figure 1c). The built-in potential generates an electric field across the junction for electrons and holes to be collected at the electrodes. Before conducting the optoelectronic measurements, we performed an assessment of the electrical properties of the Sb_2_Te_3_/MoS_2_ junction by applying a source-drain bias voltage (V_sd_) ranging from −0.75 V to 0.75 V. This evaluation allowed us to investigate the dark current behavior and assess the performance of the van der Waals p-n heterojunction. As shown in Figure 1d, the dark current remained stable and consistent across the range of applied bias voltages. Specifically, the measured dark current was found to be as low as a few picoamperes (minimum system noise floor) at zero bias and increased merely to 29 pA/23 pA at ±1 V bias. These low levels of dark current are a direct result of the unique heterostructure design, which effectively suppresses unwanted leakage currents and improves device performance, enabling high noise equivalent power (NEP). 

To evaluate the spectral response of the photodetectors, the devices are illuminated with a supercontinuum laser source. A wavelength sweep on the illuminated device was performed to observe the photocurrent across the spectrum. The laser is precisely directed towards the junction region of the photodetectors, employing a range of power levels. Specifically, these levels correspond to intensities of 0.55 μW, 2.01 μW, and 3.85 μW, respectively.

The power-dependent current–voltage (IV) characteristics measurement of the p-n heterojunction has been conducted across a range of wavelengths from 500 nm to 900 nm with a step size of 10 nm. When the bias voltage is zero, the heterojunction effectively separates the photogenerated electron-hole pairs, which then generate a photocurrent due to the built-in electric field established at the MoS_2_/Sb_2_Te_3_ interface. In Figure 2a, the relationship between the measured photocurrent (I_photo_) and the wavelength can be observed. The optical power remains the same at 0.55 μW for all the corresponding wavelengths. To distinctly discern the variation in current at a zero external bias voltage, Figure 2b delineates the regime outlined in Figure 2a, specifically ranging from −0.2 V to 0.2 V. At this zero bias point, the current exhibits a fluctuation from −0.12 nA to 0.1 nA, corresponding to the wavelengths between 500 nm and 900 nm. The photoresponsivity, represented as R, is subsequently calculated using the equation R = (I_ds_ − I_d_)/P_in_, as illustrated in Figure 2c. At zero bias, the photoresponsivity corresponding to a wavelength of 500 nm escalates to a value of 0.21 mA W^−1^. By introducing an external bias voltage of −0.2 V and 0.2 V, the photoresponsivity amplifies to respective values of 0.9 mA W^−1^ and 2.3 mA W^−1^. When the power of incident laser is kept constant, the observed variation in photocurrent by changing the wavelengths is primarily due to the energy carried by the photons. Different wavelengths correspond to photons with varying energies. In a 2D material-based photodetector, the absorption of photons and subsequent generation of photocurrent are governed by the material’s bandgap energy. When the incident light has a shorter wavelength, the photons possess higher energy and can more easily excite electrons across the bandgap, resulting in a larger photocurrent. Conversely, longer wavelengths carry lower energy, making it more difficult for the photons to promote electron transitions, leading to a comparatively smaller photocurrent. Thus, even with the same power of light, the variations in photocurrent arise from the various energy levels associated with different wavelengths of light. The depth of the dip depends on the specific properties of the semiconductor material, such as the bandgap energy and the density of states in the conduction and valence bands. As the intensity of the optical power illumination increases, a corresponding rise in the photocurrent is observed due to the amplified generation rates of the electron-hole pairs (Figure 2d). This behavior emphasizes the potential for self-powered photodetection across a broad range of wavelengths. The relationship between the incident optical power and the resulting photocurrent is often nonlinear and can be affected by factors such as the bandgap energy of the semiconductor material, the doping level, and the contact configuration of the device. However, although the increase in incident optical power will lead to a corresponding increase in photocurrent, after reaching a certain power intensity, the saturation effect or the damage of the device will undermine the responsivity performance (Figure 2e). At a +1 V bias, the responsivity experiences an amplification depending on both the wavelength and the optical power. As the applied voltage increases, the responsivity continues to grow and eventually reaches a saturation point that is contingent on the incident optical power. This saturation effect highlights the importance of understanding the optimal operating conditions for the photodetection device as it may influence the device’s overall efficiency and effectiveness in various applications. Balancing the illumination intensity and other parameters is crucial to maximizing the device’s responsivity without reaching the saturation threshold, thus ensuring optimal performance in a wide range of photodetection scenarios. 

As depicted in Figure 2f, the device exhibits a photoresponsivity of approximately 0.1 mA W^−1^ in the absence of any external bias. This finding underscores the heterojunction’s ability to effectively perform photodetection tasks without the need for additional bias voltage, highlighting its potential for ultra-low power applications. When measuring the wavelength-dependent responsivity of the photodetection device, we found that, under a bias voltage of 0.1 V, the difference in responsivity from 500 nm to 900 nm wavelengths is less than 0.1 mA W^−1^. This observation suggests that the proposed device is capable of functioning effectively across a broad photoresponsive spectrum, ranging from visible to near-infrared (NIR) regions. Moreover, the device exhibits a non-dispersive responsivity of approximately 0.3 mA W^−1^ when operating under a 0.1 V bias, demonstrating its potential for efficient broadband photodetection applications with constant responsivity over a broad wavelength range. When a positive bias voltage is applied, an external electric field is created at the junction interface, which enhances the carrier separation efficiency of the photogenerated carriers, consequently boosting the responsivity. 

To gain a deeper insight into the impact of mechanical strain on photoresponsivity, an experiment to measure the photocurrent (I_ds_) at varying levels of strain and light intensity was conducted. Initially, the I–V curves were measured under different levels of strain (0%, 0.12%, and 0.3%) in the absence of light, as shown in Figure 3a. We observe a significant strain-modulated behavior, with the rectification characteristics weakening under tensile strain conditions. To observe the alteration in current at zero bias voltage, Figure 3b presents a logarithmic scale plot of the strain-dependent I–V under dark conditions. These findings suggest that the application of mechanical strain can have a significant impact on the electrical properties of the heterojunction, leading to changes in the dark current. For instance, at a bias voltage of 0.6 V, we observed a decrease in the dark current from 14 pA to 9 pA as the tensile strain increased from 0% to 0.12%, with a further drop to 2 pA at a strain of 0.3%. This behavior can be attributed to the strain-induced changes in the device’s energy band structure. However, the signal became noisier as the current levels approached the measurement’s setup limits, with environmental noise being a significant factor. It is important to carefully control the testing environment and to use appropriate measures to reduce any sources of noise that could affect the results. 

Mechanical strain can also introduce changes in the resistance of a photodetection device. When a semiconductor material is subjected to mechanical strain, the lattice structure of the material is distorted, leading to changes in the effective cross-sectional area and the length of the material. As a result, the resistance of the device is altered due to changes in the free carrier concentration and mobility. Under tensile strain conditions, the resistance of the device generally increases due to the reduction in the doping concentration and carrier mobility. This increase in resistance can have a significant impact on the performance of the photodetection device as it can affect the magnitude of the photocurrent and the photoresponsivity of the device. Furthermore, the change in resistance can also impact the noise level of the device. The increase in resistance can lead to an increase in the Johnson noise of the device, which is generated due to the thermal agitation of the carriers in the material. This noise can contribute to the overall noise level of the device and reduce its signal-to-noise ratio. Compared to the zero-strain condition, the separation of electron-hole pairs by the built-in electric field occurs primarily in or near the heterojunction upon illumination, with electrons (holes) being swept into the MoS_2_/Sb_2_Te_3_ layer, resulting in a photocurrent generation. When a tensile strain is applied, the total internal electric field and the electric potential difference within the heterojunction are reduced, as previously discussed. This reduction hinders the separation of electron-hole pairs and impairs the injection efficiency in or near the heterojunction, leading to a decrease in both I_ph_ and responsivity when the tensile strain increases. As illustrated in Figure 3c, the overall responsivity exhibits a decreasing trend as the tensile strain exerted on the device escalates, suggesting that the ratio of photocurrent to dark current diminishes as the degradation of photocurrent surpasses that of the dark current. Figure 3d–f delve into the analysis of power-dependent responsivity alteration under three distinct tensile strain conditions: 0%, 0.12%, and 0.3%. This analysis also considers incident laser powers of 0.55 μW, 2.01 μW, and 3.85 μW. The application of an external bias to the device allows it to withstand increased strains without significant performance degradation. The strain impacts the self-powered capability of the device more substantially. Specifically, as the strain intensifies, the responsivity under zero bias experiences a decline of approximately 20 db when comparing the 0% and 0.12% strain conditions. With a bias voltage of ±1 V, the responsivity only fluctuates by about 3 db under the same comparative strain conditions.

The time response measurements of the Sb_2_Te_3_/MoS_2_ heterojunction photodetectors were performed under zero bias conditions utilizing a pulsed laser with a fixed wavelength and intensity. The resulting photocurrent was accurately captured using a high-speed oscilloscope, enabling precise determination of the rising and falling times. The obtained results indicated a rising time of 0.197 s and a falling time of 0.138 s, demonstrating rapid response and efficient carrier dynamics within the Sb_2_Te_3_/MoS_2_ heterojunction photodetector. The response time, τ, was calculated as the duration for the photocurrent to transition from 10% to 90% during the rising phase (rise time) and from 90% to 10% during the falling phase (recover time), respectively. However, it should be noted that the observed response speed is lower than initially anticipated, necessitating a deeper investigation into the underlying factors contributing to this limitation. To identify the causes behind the reduced RF response, two primary factors were examined: surface traps or defects present in the Sb_2_Te_3_ and MoS_2_ layers and interface states located at the Sb_2_Te_3_/MoS_2_ junction. Surface traps or defects have the potential to act as recombination centers, impeding carrier mobility and reducing their lifetimes. The presence of interface states can also hinder the efficient transfer of charges, ultimately degrading the overall device performance. Thorough analysis and characterization of these factors are imperative to mitigate their impact and to enhance the RF response of the Sb_2_Te_3_/MoS_2_ heterojunction photodetector.

## 4. Conclusions

In conclusion, mechanical strain significantly affects the performance of a photodetection device, with strain-induced changes in the bandgap, resistance, and depletion region all affecting the device’s photoresponsivity. Under conditions of tensile strain, the bandgap and resistance increase while the doping concentration and carrier mobility decrease, collectively resulting in a decrease in photoresponsivity. The results have significant implications for the advancement of photodetection devices, particularly in applications where mechanical strain may be present, such as wearable devices and flexible electronics. Through a thorough understanding of the impact of mechanical strain on photodetection device performance, we have discovered that applying an increased external bias voltage can enable these devices to endure greater strains. This insight paves the way for the development of more robust and reliable devices. These enhanced devices are expected to function effectively under a broad spectrum of strain conditions, thus offering greater versatility in practical applications.

## Figures and Tables

**Figure 1 nanomaterials-13-01973-f001:**
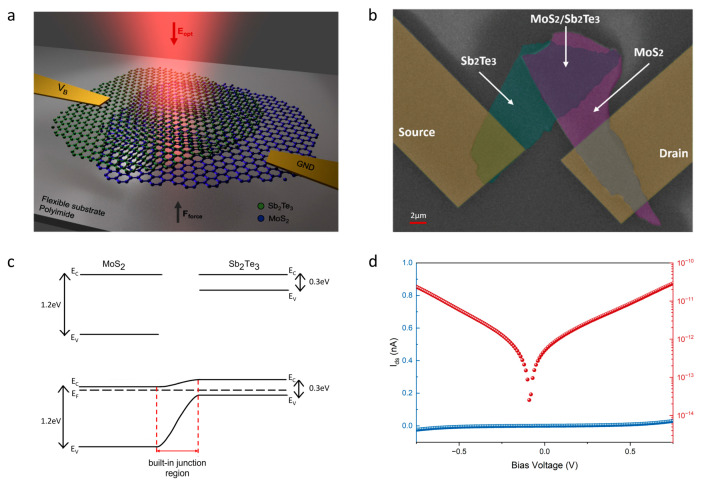
Sb_2_Te_3_/MoS_2_ heterostructure photodetector. (**a**) A schematic representation of the Sb_2_Te_3_/MoS_2_ van der Waals heterojunction photodetector. (**b**) The false colored SEM image of the Sb_2_Te_3_/MoS_2_ heterostructure device. (**c**) The band structures of the vdW layered MoS_2_/Sb_2_Te_3_ heterojunction. (**d**) The I–V characteristics of the Sb_2_Te_3_/MoS_2_ van der Waals heterojunction under −0.75 to 0.75 V bias voltage in a dark environment. The blue curve illustrates the linear plot, while the red curve represents the log scales.

**Figure 2 nanomaterials-13-01973-f002:**
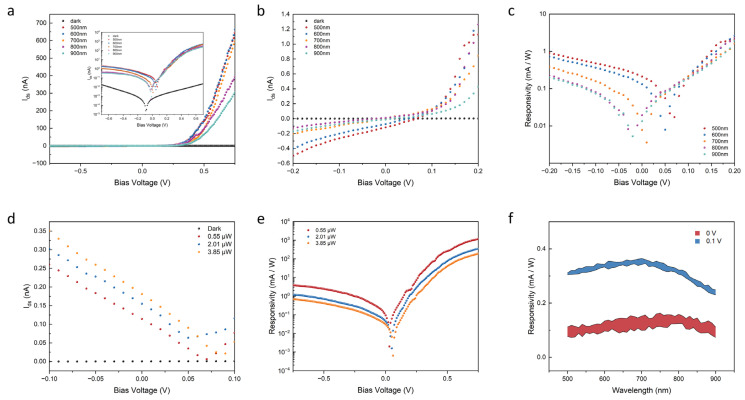
Photovoltaic characteristics of the Sb_2_Te_3_/MoS_2_ van der Waals heterojunction. (**a**) The I–V characteristics of the Sb_2_Te_3_/MoS_2_ heterojunction under different optical wavelengths with the same illumination power of 0.55 μW. (**b**) Delineates the regime outlined in the I–V characteristics, specifically ranging from −0.2 V to 0.2 V. (**c**) The measured photoresponsivity at different wavelengths (500 nm, 600 nm, 700 nm, 800 nm, and 900 nm), with the same illumination power of 0.55 μW under −0.2 to 0.2 V bias voltage. (**d**) The measured photocurrent at 500 nm in three different power levels: 0.55 μW, 2.01 μW, and 3.85 μW, respectively. (**e**) The measured photoresponsivity at 500 nm in three different power levels: 0.55 μW, 2.01 μW, and 3.85 μW, respectively. Photoresponsivity was observed and increased proportionally to the optical power under −0.75 to 0.75 V bias voltage. (**f**) Responsivity by sweeping the wavelength under 0 and 0.1 V bias from 500 to 900 nm with a step size of 10 nm, which indicates a broad wavelength response.

**Figure 3 nanomaterials-13-01973-f003:**
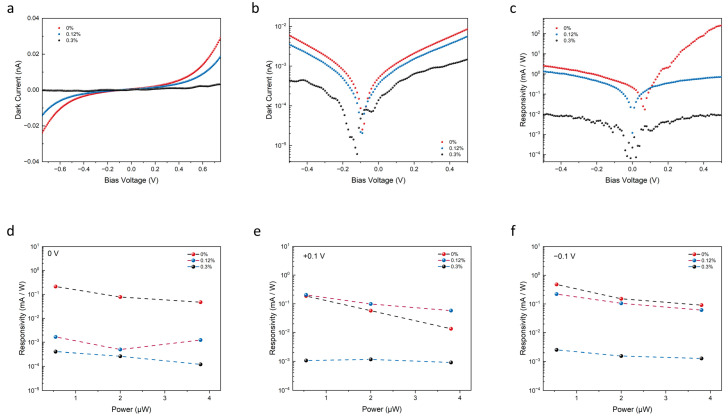
Strain-dependent photoresponse of the Sb_2_Te_3_/MoS_2_ van der Waals heterojunction. (**a**,**b**) The I–V characteristic of the device was measured under varying strains without illumination on a flexible substrate. The dark current decreased substantially as the tensile strain increased. (**c**) The photoresponsivity of the device was measured under different tensile strains at 500 nm wavelength illumination. As the strain increased, the responsivity decreased, indicating a simultaneous drop in both the photo and dark currents. However, the photocurrent exhibited a more significant decrease. (**d**–**f**) The power-dependent responsivity change of the heterojunction device was measured under different strains with a 500 nm laser illumination at 0 V, 0.1 V, and −0.1 V.

## Data Availability

The data that support the findings of this study are available from the corresponding author upon reasonable request.

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
