# Peer review of "Self-Powered Sb2Te3/MoS2 Heterojunction Broadband Photodetector on Flexible Substrate from Visible to Near Infrared"

_nanomaterials, 2023, doi:10.3390/nano13131973_

Round 1
Reviewer 1 Report
This study demonstrated self-powered photodetector with two different 2D materials. Interestingly, low-power operation is observed. If the author addresses following comments, it would be enough to be published.
1 This study is about 2D TMD semiconductor-based photodetector. The title should include “2D TMD crystal materials”.
2 In figure 1c, the built-in junction region should be indicated in the band diagram.
3 Data figures should be larger for readers to see clear. Especially, it is very difficult to recognize inset of Figure 2a.
4 Figure 2a shows I-V with various light wavelengths. Different photocurrent is changed by changing wavelengths. According to “photoelectric effect” theory, photo current is changed by light intensity and not by light energy(wavelength). Please describe about this in the manuscript.
5 Figure 2d shows the photocurrent with various light intensity. Power-law data can be achieved with log(Iph) = Alog(P). Iph is photocurrent, P is light power, and A is power-law exponent. Please check this reference [Kind et al. Nanowire Ultraviolet Photodetectors and Optical Switches, Advanced Materials, vol 14, p 158, (2002)].
6 Switching speed is also an important parameter to see the performances. Is it possible to measure time-dependent switching behavior?
Reviewer 2 Report
1. Please rewrite the abstract with more specific information such as name of sensing material, response/recovery time, detectivity, responsivity, etc.
2. Line 32, How can a photodetector be used for energy harvesting? 3. Line 58, Please include the specific names of TMDs. 4. Explain the method for electrodes deposition. 5. Please include the SEM images for the developed heterostructures. 6. Author may include the information for the thickness of exfoliated MoS2 and Sb2Te3 nanostructures. 7. The detailed information about the heterostructure construction must be a part of the manuscript. 8. Almost 17 references belong to the authors of the manuscript. Self-citation must be reduced. 9. A comparison table for comparing the results with the latest literature survey can be included in the manuscript. Author can compare the results with following articles: (i) https://doi.org/10.1016/j.jmrt.2021.07.032 (ii) https://doi.org/10.3390/nano12030325Author Response
Please see the attachment

Reviewer 3 Report
REVIEW
on the manuscript “Self-powered Broadband Photodetector on Flexible Substrate from Visible to Near Infrared Wavelength”
by Hao Wang, Chaobo Dong, Yaliang Gui, Jiachi Ye, Salem Altaleb, Martin Thomaschewski, Behrouz Movahhed Nouri, Chandraman Patil, Hamed Dalir, Volker J Sorger
In the presented manuscript the results of experimental investigations of the photodetectors of visible and infrared range based on van der Waals heterostructures with 2D MoS2/Sb2Te3 p-n heterojunction are presented. Current-voltage characteristics under dark and light illumination and photoresponsivity of the proposed photodetectors are analyzed. The influence of compressive strain on the parameters of photodetector is studied. These structures and technological approach to the device design should be useful for further device applications in optoelectronics.
Chosen methods are adequate and up-to-date, while results of experimental research are reliable. Overall, the paper is organized and written very well and at high scientific level.
However, in the reviewer’s opinion, the manuscript should be slightly improved before publication. The details are listed below.
1. Abstract, Line 25: Please remove the duplicated word “applications”.
2. The PIC abbreviation in the keywords should be deciphered.
3. Introduction section, Line 67: What are the non-linear coefficients mentioned in this sentence. Please clarify.
4. What is the meaning of the blue circle curve in Figure 1d. The values are so small that they are not visible. The same is true for all other analogous curves.
5. The inset to Figure 2a is not readable. Maybe it is better to make a separate figure from it.
6. Capture to Figure 2: The phrase “The I–V characteristics of the Sb2Te3/MoS2 heterojunction under different wavelength optical with the same illumination power” is confusing. Please correct.
7. Figure 2f: What is the minimum of responsivity at the wavelength of 600 nm may be accounted for? And why it is pronounced only at zero bias?
8. Comparison of the obtained values of photodetector parameters with some other photodetectors from the literature may be carried out.
9. Additionally, the language of manuscript should be double-checked and some other typos are removed, for example in Line 38.
Conclusion: The presented manuscript may be published in the Nanomaterials journal after moderate revision.
Minor editing of English language required.
Round 2
Reviewer 2 Report
I am doubtful about figure 1(b). Is it really a scanning electron microscope (SEM) image?
Rest comments are addressed properly. Article may be accepted now.
